# Optimization of the Guiding Stability of a Horizontal Axis HTS ZFC Radial Levitation Bearing

António J. Arsénio [1,2,*], Francisco Ferreira da Silva [1,2], João F. P. Fernandes [1,2] and Paulo J. Costa Branco [1,2]

1   Instituto de Engenharia Mecânica (IDMEC), Av. Rovisco Pais, 1, 1049-001 Lisboa, Portugal; francisco.ferreira.silva@tecnico.ulisboa.pt (F.F.d.S.); joao.f.p.fernandes@tecnico.ulisboa.pt (J.F.P.F.); pbranco@tecnico.ulisboa.pt (P.J.C.B.)
2   Instituto Superior Técnico (IST), Universidade de Lisboa, 1049-001 Lisboa, Portugal
*   Correspondence: antoniojcosta@tecnico.ulisboa.pt

**Abstract:** This document presents a study on the optimization of the 3D geometry of a horizontal axis radial levitation bearing with zero-field cooled (ZFC) high-temperature superconductor (HTS) bulks in the stator, and radially magnetized permanent magnet (PM) rings in the rotor. The optimization of component dimensions and spacing to minimize the volume or cost concerning only the maximization of the levitation force was previously studied. The guidance force and guiding stability depend on the spacing between PM rings in the rotor and between the rings of HTS bulks in the stator. This new optimization study aims to find the optimum spacing that maximize the guidance force with given HTS bulk and PM ring dimensions while maintaining the minimum required levitation force. Decisions are taken using the non-dominated sorting genetic algorithm (NSGA-II) over 3D finite element analysis (FEA). A simplified electromagnetic model of equivalent relative permeability is used on 3D FEA to reduce numerical processing and optimization time. Experimental prototypes were built to measure magnetic forces and validate appropriate values of equivalent magnetic permeability. An analysis of stable and unstable geometry domains depending on the spacing between rings of HTS bulks and PM rings is also done for two HTS bulk sizes.

**Keywords:** geometry optimization; guiding stability; high-temperature superconductor; radial levitation bearing; zero-field cooling

## 1. Introduction

The studied horizontal axis radial levitation bearing comprises a stator with two liquid nitrogen (LN$_2$) chambers for housing two discontinuous rings of zero-field cooled (ZFC) high-temperature superconductor (HTS) bulks and a permanent magnet (PM) rotor with three radially magnetized PM rings in an alternating polarization arrangement, each HTS ring being positioned in the zone between adjacent PM rings.

With a uniform distribution of HTS bulks around the stator chambers, the magnetic levitation force is zero with the rotor and stator axes aligned (no vertical deviation of the rotor). Due to the rotor weight, it is not possible to keep the rotor levitating at the center position in this situation. To maximize the available net levitation force (difference between levitation force and the rotor weight) and the range of rotor vertical deviations with a positive value of net levitation force, a topology with only six HTS bulks at the bottom of the stator was considered in experimental measurements for the validation of electromagnetic model parameters. Figure 1a shows the tested bearing topology with six HTS bulks, Figure 1b shows the exploded 3D view of the experimental prototype design, and Figure 1c the 3D perspective of the bearing assembly.

In the experimental prototypes, the stator with two LN$_2$ chambers of rigid high-density polyurethane (PUR) walls housed one discontinuous ring of HTS bulks. The stator walls were built by a computer numeric control (CNC) milling machine. The rotor structure

in the polylactic acid plastic (PLA) was printed by a 3D computer-aided design (CAD) printer [1].

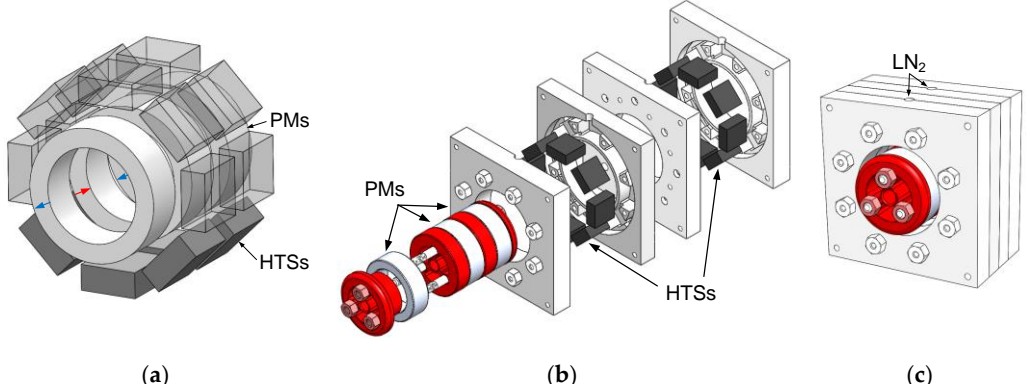

(**a**)                (**b**)                (**c**)

**Figure 1.** (**a**) Tested bearing topology with six HTS bulks, (**b**) 3D exploded view, and (**c**) 3D perspective of the experimental prototype assembly.

Initially, a first stator, referred to as Stator I, for housing bulks with dimensions $33 \times 33 \times 14$ mm$^3$ as built. This stator and two rotors with 5 mm and 20 mm spacing between PM rings, named rotors D5 and D20, were used to measure magnetic levitation and guidance forces and validate electromagnetic model parameters [2].

Studies on the LN$_2$ consumption and YBCO bulk temperature evolution for different thermal processes in the initial ZFC and the operation of the experimental bearing prototype were presented in [3,4].

Work on optimizing the superconducting linear magnetic bearing of a maglev vehicle was presented [5]. A multiobjective multi-constraint optimization to minimize the cost or volume of the 3D bearing geometry, considering variable component dimensions and spacing, was initially performed in [6]. Decisions were based on the non-dominated sorting genetic decision algorithm NSGA-II over 3D finite element analysis (FEA) results. A simplified model of equivalent relative permeability was adopted to reduce the 3D FEA numerical processing. This initial optimization study considered only the maximization of levitation forces, not looking for the maximization of the guiding stability. A minimum levitation force equal to the one obtained with the geometry defined by six HTS bulks at the bottom of Stator I and rotor D5 was considered a restriction. Figure 2 compares this restriction geometry with the resulting volume and cost-optimized geometries.

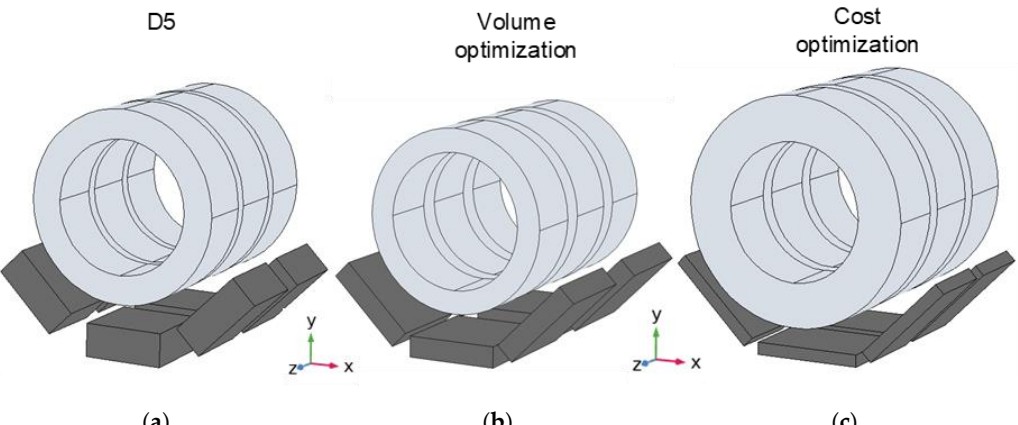

(**a**)                (**b**)                (**c**)

**Figure 2.** (**a**) Geometry with rotor D5 and six HTS bulks of $33 \times 33 \times 14$ mm$^3$, (**b**) Volume, and (**c**) Cost-optimized geometries.

Another stator referred to as Stator II, for housing bulks with dimensions $40 \times 40 \times 10$ mm$^3$, close to the bulk dimensions in the volume-optimized geometry was built. Figure 3 shows the chamber profiles of Stator I and Stator II.

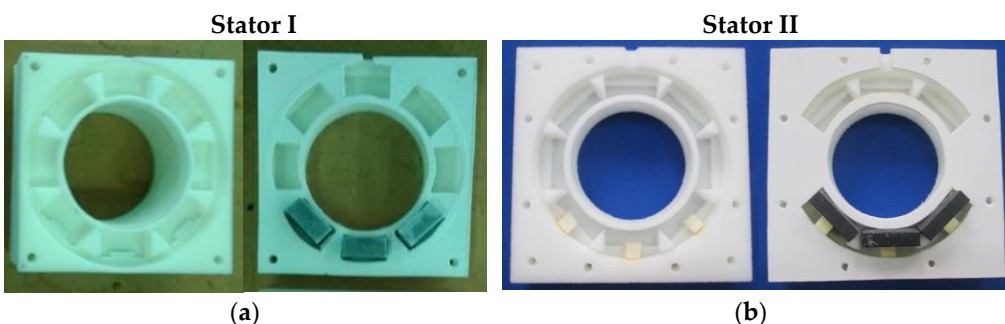

**Figure 3.** Chamber profiles of (**a**) Stator I for bulks with a volume $33 \times 33 \times 14$ mm$^3$ and (**b**) Stator II for bulks with a volume $40 \times 40 \times 10$ mm$^3$.

The studied bearing is a passive magnetic bearing (PMB), with the magnetic field generated by the rotor's three radially magnetized PM rings. The polarization of the middle PM ring is opposite from the polarization of the side ones. PM rings of neodymium iron boron (NdFeB) of grade N40 with a remanent magnetic flux density $B_r = 1.25$ T and HTS bulks of yttrium barium copper oxide (YBCO), fabricated in a top-seeded melt growth (TSMG) process, were used in experimental prototypes. Table 1 states electrical and mechanical parameters and the main geometric dimensions for the tested bearing experimental prototypes.

Most of the levitation vehicles and bearings with superconducting bulks adopt FC to have stability by flux pinning. The studies in [7,8] compare impulsion forces obtained using ZFC with the ones obtained by FC at several cooling heights, showing that the levitation forces with ZFC are higher. Results on levitation or guidance forces, using ZFC or FC at several heights, for an HTS Maglev launch assist test vehicle, including a guideway with three alternate polarization PM lines and HTS bulks, were presented in [9]. The optimization of the geometry of a radial levitation bearing with a similar arrangement of PMs and with one continuous HTS ring cooled by FC or ZFC was carried out in [10,11].

With ZFC, the magnetization energy and Joule losses in the bulks are minimized, because of no initial flux pinning, increasing their lifetime. Levitation forces are higher than with FC at several heights, as shown in [7,8]. Because there is no flux pinning with ZFC, guiding stability is only guaranteed by geometries with a specific arrangement of PMs and spatial distribution of HTS bulks. Thus, it is of extreme importance to determine which geometries present guiding stability.

The studied frictionless bearing can be applied in high-speed applications. Induction currents, due to dynamics, could appear even with ZFC. Active control to generate compensation forces and reduce vibrations and losses could be necessary for high-speed applications. Reference [12] showed that, in high-speed applications, Samarium Cobalt (SmCo) is an alternative of NdFeB for PMs because induction current losses could be reduced with the first alloy. A study on optimizing an axial flux PM machine for torque ripple minimization is presented in [13]. A process to control the displacement and speed of a motor with two radial levitations and one axial thrust magnetic bearing was described in [14]. The generation of compensation forces to actively control the displacement of a flywheel with passive and hybrid magnetic bearings is described in [15].

The guiding stability can be measured by the maximum available energy to pull back the rotor to the equilibrium position, calculated by the product between the maximum guiding force times the corresponding rotor axial deviation or by the axial or guiding stiffness. A study on the optimization of the spacing between PMs and between HTS bulks to maximize the guiding stability for the proposed ZFC configuration is necessary to complement and complete the research on the studied HTS ZFC levitation bearing.

This document presents a study on optimizing the spacing between HTS bulk rings in the stator and between PM rings in the rotor, to maximize the guidance force with a restriction on the minimum levitation force that should be guaranteed. The domain of

the combination of spacing for which there is guiding stability is also determined for two different bulk sizes.

**Table 1.** Electrical and mechanical parameters and main geometric dimensions for the tested bearing experimental prototypes.

| | | |
|---|---|---|
| **Stator** | Number of HTS rings & LN$_2$ chambers | Two |
| | Maximum number bulks per chamber | Eight uniformly distributed |
| | Number of bulks in tested topologies | Three at the bottom of each chamber |
| | Height and width of the exterior surface | 170 mm |
| | Total length with two LN$_2$ chambers | 100 mm for Stator I 108 mm for Stator II |
| | Volume of each LN$_2$ chamber | 362 cm$^3$ |
| | Diameter of the rotor cavity | 90 mm |
| | Material of chamber walls | Rigid polyurethane of 40 kg m$^{-3}$ |
| | Composite of HTS bulks | YBa$_2$C$_3$O$_7$ crystal by TSMG |
| | Size of HTS bulks | $33 \times 33 \times 14$ mm$^3$ in Stator I $40 \times 40 \times 10$ mm$^3$ in Stator II |
| **Rotor** | Number of PM rings | Three |
| | Direction of magnetization | Radial |
| | Arrangement of polarizations | Alternate parallel ↑↓↑ |
| | Outer diameter of rotor and PM rings | 79 mm |
| | Inner diameter of PM rings | 55 mm |
| | Material of rotor structure | 3D printed PLA |
| | Composite of PMs | NdFeB of grade N40 ($B_r = 1.25$ T) |
| | Length of rotors | 100 mm for Rotor D5 130 mm for Rotor D20 |
| | Weight and Gravity force | 1.57 kg & 15.39 N for Rotor D5 1.62 kg & 15.88 N for Rotor D20 |
| | Momentum of inertia | $1.59 \times 10^{-3}$ kg m$^2$ for Rotor D5 $1.64 \times 10^{-3}$ kg m$^2$ for Rotor D20 |

## 2. Electromagnetic Models and Parameters

The E-J model described below should be used for a detailed prediction and characterization of the induced current distribution in HTS bulks. This model implies the resolution of non-linear and partial differential equations in the superconductor domain, requiring a lot of numerical processing in the FEA. A simplified model with a calibrated value of equivalent relative permeability was used to significantly reduce the FEA processing time, especially during the optimization of the bearing 3D geometry.

### 2.1. E-J Model

When, after ZFC, a magnetic field **H**$_a$ is applied to a superconductor bulk, an electric field **E** is induced by the variation of the penetrating magnetic field **H**, according to Faraday's law.

$$\nabla \times \mathbf{E} = -\mu_0 \frac{d\mathbf{H}}{dt}. \tag{1}$$

According to the model in [16], type II superconductors present a non-linear electric conductivity characteristic given by the power-law (2),

$$E = E_0 \left( \frac{J}{J_c} \right)^n,$$ (2)

where $E$ and $J$ are the magnitudes, respectively, of induced electric field and current density, $J_c$ is the critical current density and $E_0$ is the electrical field at which the current density reaches $J_c$. Exponent $n$ is a positive integer higher than 1. The induced current density **J** creates, by Ampere's law, a magnetization field **M**.

$$\mathbf{J} = \nabla \times \mathbf{M}.$$ (3)

The penetrating magnetic field is given by the sum of the applied magnetic field and the magnetization field.

$$\mathbf{H} = \mathbf{H}_a + \mathbf{M}.$$ (4)

According to the Kim–Anderson model in [17,18], the critical current density $J_c$ depends on the absolute value of the penetrating magnetic-flux density $|B|$,

$$J_c(B) = J_{c0} \frac{B_0}{B_0 + |B|},$$ (5)

where $J_{c0}$ is the zero-field critical current density that depends on the temperature $T$ [19], and $B_0$ the magnitude of the penetrating magnetic flux density for which the critical current density is half the zero-field critical current density. For the case of yttrium barium copper oxide (YBCO), $E_0 = 1 \times 10^{-4}$ Vm$^{-1}$, $B_0 = 0.1$ T and $n = 21$ [20]. Appropriate values of the parameter $J_{c0}$ can be validated by comparing the magnetic forces predicted by FEA using the E-J model with the ones obtained by experimental measurement.

### 2.2. Equivalent Relative Permeability Model

The relative permeability $\mu_r$ is defined by the relation between the magnitudes of the penetrating field $H$ and applied magnetic field $H_a$ and is calculated by (6).

$$\mu_r = \frac{H}{H_a} = 1 + \frac{M}{H_a} = 1 + \chi,$$ (6)

where $\chi$ is the magnetic susceptibility given by the relation between the magnitudes of the magnetization $M$ and the applied magnetic field $H_a$.

In this model, an average value of relative permeability $\overline{\mu_r}$ designated by the equivalent relative permeability is considered. A methodology to compute the value of $\overline{\mu_r}$ for the total bulk volume was initially proposed in [2,6]. Based on the first methodology proposed, a more generic methodology to compute values of $\overline{\mu_r}$ in several partitions of the bulk was then proposed in [21].

Appropriate values of $\overline{\mu_r}$ can be validated by comparing the magnetic forces predicted by FEA with the ones obtained by experimental measurement. The use of this model significantly reduces the FEA numerical processing.

### 3. Validation of Electromagnetic Model Parameters

Magnetic levitation and guidance forces were measured experimentally and were determined by 3D FEA for two geometries, respectively, with rotors D20 and D5 and with six YBCO bulks of volume $33 \times 33 \times 14$ mm$^3$ at the bottom of Stator I. In these two geometries, the spacing between the two YBCO bulk rings was 10 mm. Figure 4a,b show the dimensioned 3D perspective design of the half part used in FEA simulations for these two geometries.

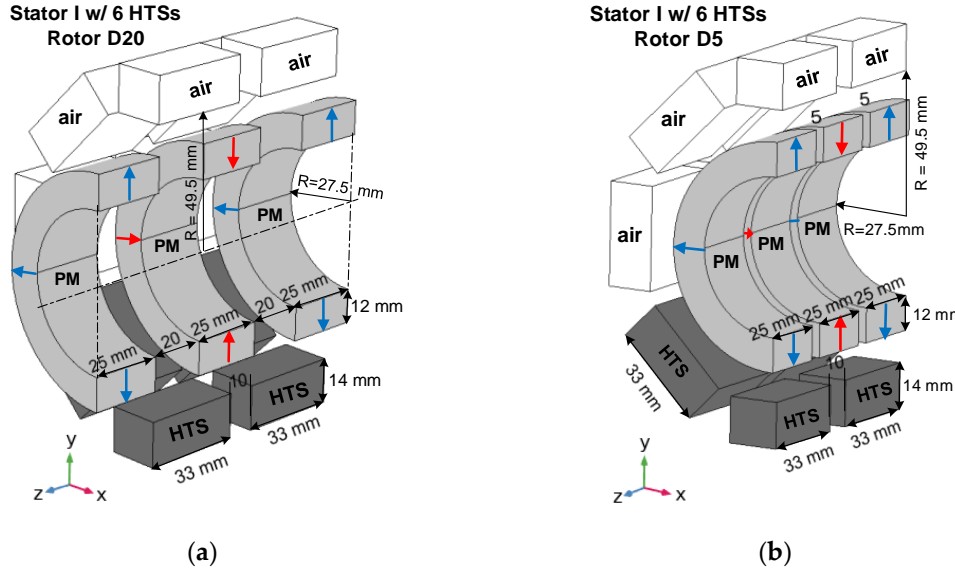

**Figure 4.** The dimensioned 3D perspective of the half part used in FEA simulations for the two tested geometries with six HTS bulks at the bottom of stator I and with rotors (**a**) D20 or (**b**) D5.

Figure 5a,b show, respectively, the transversal and longitudinal views of the magnetic flux and current density distributions for the geometry with rotor D20. Figure 6a,b refer to the geometry with rotor D5. The results were obtained using the E-J model with $J_{c0} = 8 \times 10^7$ Am$^{-2}$.

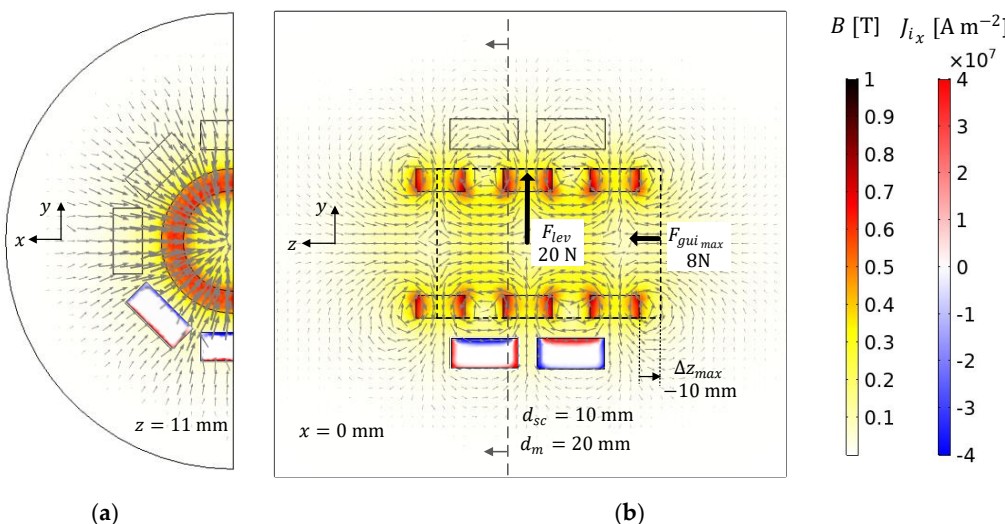

**Figure 5.** (**a**) Transversal and (**b**) longitudinal views of the magnetic flux and current density distributions for the geometry with six HTS bulks at the bottom of Stator I and rotor D20.

Experimental measurements and FEA simulations were performed to verify the dependence of the levitation force $F_{lev}$ with the rotor vertical deviation and the dependence of the guidance force $F_{gui}$ with the rotor axial deviation keeping the rotor and stator axes aligned. Diagrams showing vectors of the levitation force $F_{lev}$ with the rotor and stator axes aligned and no rotor axial deviation and the maximum guidance force $F_{gui_{max}}$ obtained by pulling the rotor with a negative axial deviation, keeping the rotor and stator axes aligned, are represented in Figures 5b and 6b. The dashed lines represent the rotor position for which the guidance force magnitude is maximum. The corresponding axial deviation is given by $\Delta z_{max}$.

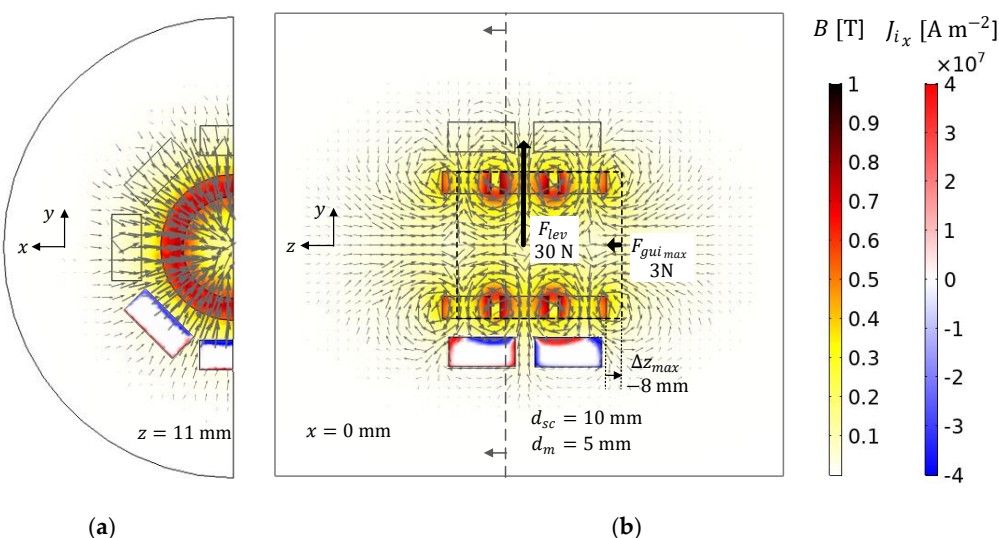

**Figure 6.** (**a**) Transversal and (**b**) longitudinal views of the magnetic flux and current density distributions for the geometry with six HTS bulks at the bottom of Stator I and rotor D5.

In the 3D FEA, a fine triangular mesh with a maximum element size of 2 mm in bulks, 3 mm in PMs rings, and 6 mm in the surrounding space were used. A linear shape function to minimize processing time was adopted. In [22], it was shown that, with linear shape functions and mesh size, there is good accuracy for the forces predicted by 3D, compared with experimental values. The characteristics of the force versus the rotor deviation, obtained using the E-J model, follow the monotony of the characteristics obtained experimentally. The 3D FEA was performed with the said shape functions and mesh grid definitions and using the software in [23].

Figure 7a,b show photos of rotors D20 and D5, indicating the gravity force $F_g$ of their weights. Figure 7c shows a photo of Stator I, with a fixed spacing of 10 mm between the two rings of HTS bulks. Figure 7d,e show the setups used to measure the levitation forces with the vertical deviation and the guidance forces with the axial deviation.

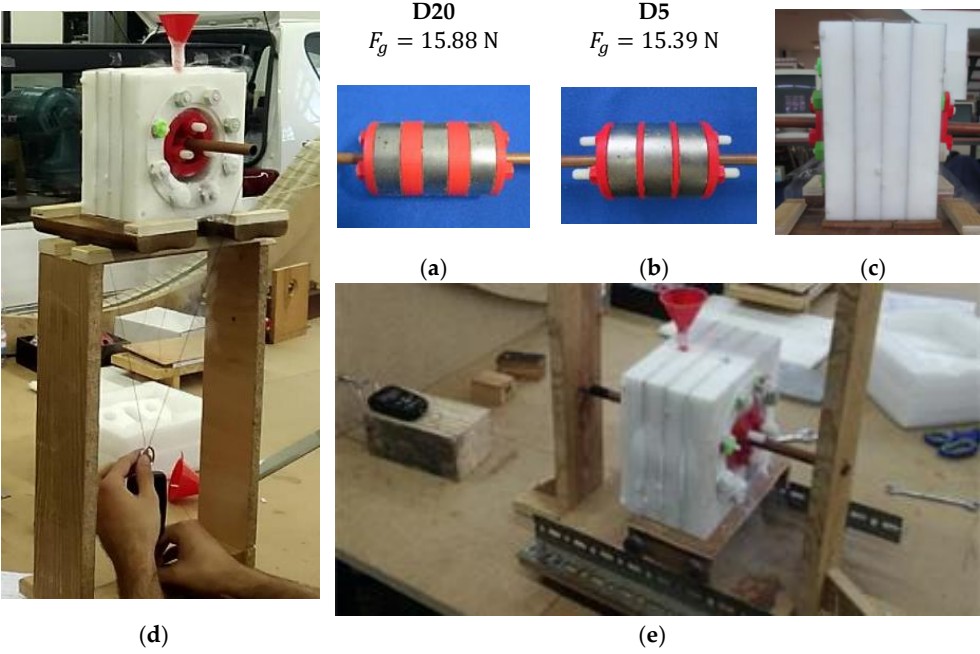

**Figure 7.** (**a**) Rotor D20, (**b**) Rotor D5, (**c**) Stator I with a spacing of 10 mm between HTS rings, and experimental setups to measure (**d**) levitation and (**e**) guidance forces.

Forces were measured using the digital dynamometer *Baxtran KRN5* with a resolution of 1 g and a maximum capacity of 5 kg. The dynamometer was on one side with its rotating ring attached to the tensor wires pulling the rotor against the measured force, on the other side with its non-rotating hook fitted to a ring with a screw. During the measurement of the net levitation force, the vertical displacement was set by the number of turns of the crew into a nut embedded at the bottom part of the support structure, as shown in Figure 7d.

For the measurement of the guidance force, the rotor vein was fixed and kept at a certain height aligned with the stator axis, with the vein edges fitted into the vertical structure bars. The stator was fixed onto a car with wheels rolling on and guided by the rails of a track. The same dynamometer was used. In this case, the tensor wires were attached to the car pulling the stator in the positive axial direction, traduced by an axial negative deviation of the rotor in the *z*-axis. The dynamometer hook was attached to the ring with the screw and the rotating ring attached to the tensor wires. The axial displacement was set by the number of turns of the crew into a nut embedded at a vertical board wall, as shown in Figure 7e.

The experimental measurement of forces with a dynamometer and a tensor wire enabled placing the non-cryogenic dynamometer away from the stator, avoiding its malfunctioning and damage by freezing. For a specific rotor's vertical or axial deviation, a minimum of three levitation or guidance force measurements were done considering the mean value. Each reading was done after the rotor stopped oscillating relatively to the stator and the dynamometer display stopped scanning, showing a fixed measurement.

The characteristics of the levitation force with the vertical deviation, obtained by 3D FEA using the equivalent permeability model and the E-J model and by experimental measurement respectively with Rotor D20 and Rotor D5, are shown in Figure 8a,b. The 3D FEA simulation time was between 1.5 h and 6 h with the E-J model and less than 30 s with the equivalent permeability model for each rotor deviation.

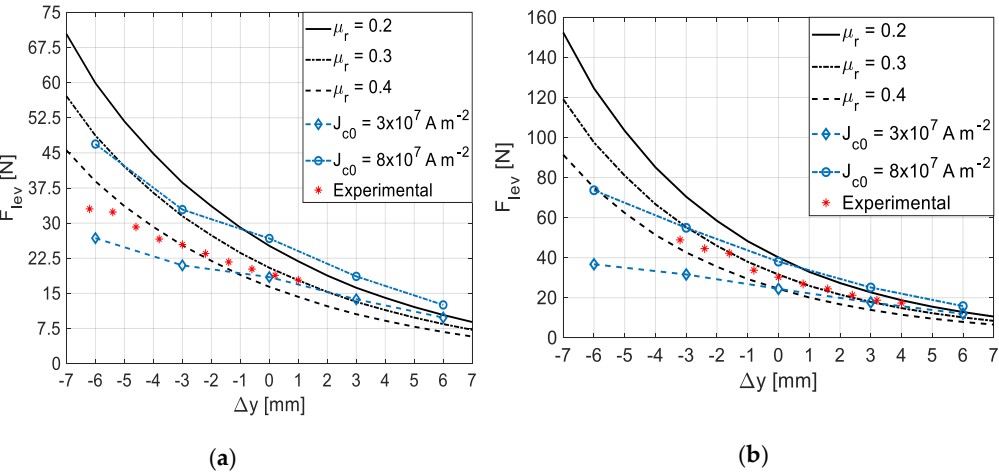

(**a**)            (**b**)

**Figure 8.** Levitation force with vertical deviation by 3D FEA using the equivalent permeability model or the E-J model and experimental measurement, with (**a**) Rotor D20 and (**b**) Rotor D5.

As one may verify, the experimental values follow the monotony of the characteristics with the E-J model between $J_{c0} = 3 \times 10^7$ Am$^{-2}$ and $J_{c0} = 8 \times 10^7$ Am$^{-2}$. For the second case with rotor D5, these are closer to the characteristic $J_{c0} = 8 \times 10^7$ Am$^{-2}$. With Rotor D5 at the center position (no vertical deviation), the levitation force with $J_{c0} = 8 \times 10^7$ Am$^{-2}$ is between the ones predicted with $\overline{\mu_r} = 0.2$ and $\overline{\mu_r} = 0.3$, that is, close to the one that would be predicted with $\overline{\mu_r} = 0.25$. The lower the distance $g_a$ between the HTSs and the PMs, the higher is the value of equivalent relative permeability for which the predicted levitation forces approximates experimental values. This is because the closer the PMs are to the HTSs, the higher the penetrating field.

Figure 9a,b show the characteristics of the levitation force with the rotor axial deviation obtained by 3D FEA using the equivalent permeability model and the E-J model and by

experimental measurement, respectively, with Rotor D20 and Rotor D5. As one may verify, the sensitivity of the guidance force to the value of $\overline{\mu_r}$ is much less than the one verified for the case of the levitation force. The applied magnetic field did not change notably in the range of axial deviations applied with the rotor at the center position.

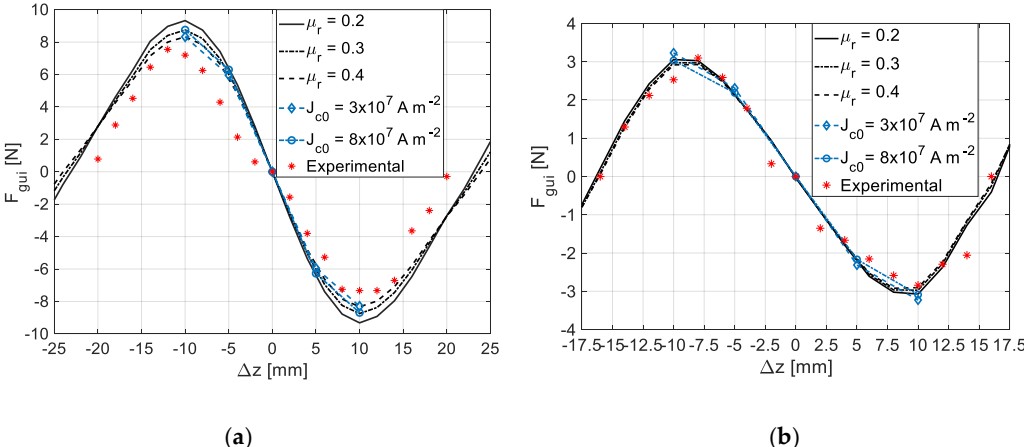

(a)　　　　　　　　　　　　　　　　　　(b)

**Figure 9.** Guidance force with axial deviation by 3D FEA using the equivalent permeability and E-J models and experimental measurement, using (**a**) Rotor D20 and (**b**) Rotor D5.

Table 2 shows the values of levitation force $F_{lev}$ with the rotor at the center position, predicted by FEA and obtained by experimental measurement. The last column presents the net force $F_n$ (difference between levitation force $F_{lev}$ and the gravity force $F_g$ of the rotor weight) relative to the experimental measurement. The errors $\varepsilon$ of the predicted forces with relation to the ones measured experimentally are also presented.

**Table 2.** Experimental and FEA values of with Rotor D20 and D5 at the center position.

| $\Delta x = \Delta_y = 0$ mm $\Delta z = 0$ mm | **Predicted Forces $F_{lev}$ [N] and Error $\varepsilon$[% ] from Measured Forces** | | | | | | | $F_n$ [N] Exp. |
|---|---|---|---|---|---|---|---|---|
| | $\mu_r = 0.2$ | $\mu_r = 0.25$ | $\mu_r = 0.3$ | $\mu_r = 0.4$ | $J_{c0} = 3 \times 10^7$ Am$^{-2}$ | $J_{c0} = 8 \times 10^7$ Am$^{-2}$ | Exp. | |
| Rotor D20 $F_g = 15.88$ N $g_a = 10.5$ mm | 25.15 +31% | 22.72 +18.4% | **20.47** **+6.67%** | 16.40 −14.5% | **18.47** **−3.75%** | 23.73 +23.7% | **19.19** | 3.31 |
| Rotor D5 $F_g = 15.39$ N $g_a = 10.5$ mm | 37.62 +24.3% | **31.39** **+3.73%** | 26.17 −13.5% | 21.44 −29.1% | 22.42 −25.9% | **32.31** **+6.8%** | **30.26** | 14.87 |

With the rotor and stator axes aligned, the levitation force predicted with $J_{c0} = 8 \times 10^7$ Am$^{-2}$ is close to the one predicted with $\overline{\mu_r} = 0.25$. The experimental values with Rotor D5 are close to the ones predicted with $J_{c0} = 8 \times 10^7$ Am$^{-2}$ and $\overline{\mu_r} = 0.25$. It was considered to select only one value of $\overline{\mu_r}$ to represent both rotors D5 and D20. The rotor D5 experiments were made first when the bulks and the PUR container were in the best conditions. After some use, the thermal insulation of the PUR wall may slightly reduce, and the HTS bulk may lower its critical current. Therefore, a value of equivalent relative permeability $\overline{\mu_r} = 0.25$ was selected to predict forces during the optimization process with the rotor and stator axes aligned.

## 4. Characterization of Dynamics for the Configuration with Six Bulks at the Bottom of Stator I

Figure 10a,b show the experimental setups to measure the dynamic responses respectively to vertical and axial deviations of PM rotors.

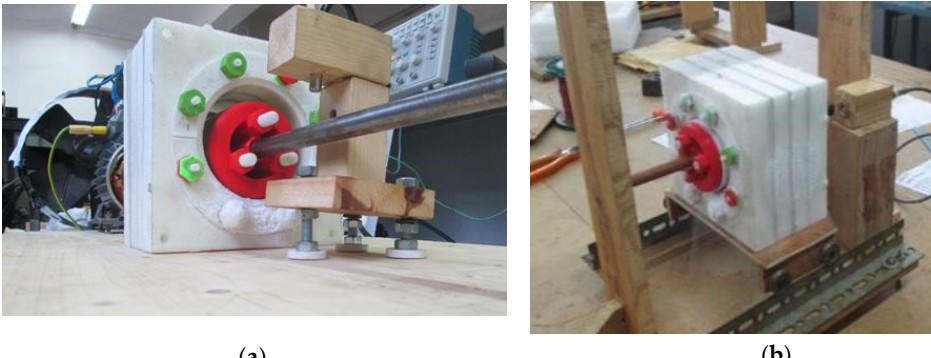

|             |             |
|:-----------:|:-----------:|
| (**a**)     | (**b**)     |

**Figure 10.** Experimental measurement of (**a**) vertical dynamics using Rotor D5 and (**b**) axial dynamics using Rotor D20.

The measurement of vertical dynamics was performed using Rotor D5 because, with this rotor, the levitation force is higher than with Rotor D20. The experimental bearing prototype guided the edge of a vein driven by a three-phase, two-pole, 375 W induction motor with a modified short-circuit rotor. A vertical deviation $\Delta y = -8$ mm was forced by pressing the shaft about 8 mm towards the negative direction of the $y$-axis until the cylindrical rotor surface touched the surface of the stator tubular cavity. At this position, the shaft was released, and the evolution of the vertical shaft position was measured with the optical sensor *PHILTEC RC190*.

On the other hand, axial dynamics were measured using Rotor D20 because its axial force is much higher than with Rotor D5. The structure used to analyze the axial dynamics was the same as the one used to measure the guidance forces $F_{gui}$ versus the axial deviation $\Delta z$, with the rotor and stator axes aligned. In this case, the ultrasonic position sensor *Baumer UNAM 12U9914/S14D* was used to measure the evolution of the coordinate $z$ from the stator transversal face. An axial deviation $\Delta z = -10$ mm was forced by moving the car carrying Stator I about 10 mm towards the positive direction of the $z$-axis. As verified in Section 3, with this axial deviation, the magnitude of the guidance force is the maximum. At this position, the car was released, measuring the evolution of the stator axial position with the ultrasonic position sensor.

Figure 11a,c show the evolutions, respectively, of the rotor vertical deviation and the levitation force in response to the stepped vertical deviation $\Delta y = -8$ mm. The evolutions of Figure 11c were determined from the ones in Figure 11a, using functions fitting the experimental characteristic presented in Figure 8b. The evolutions obtained from experimental measurements are shown in continuous blue lines. The dotted red line corresponds to the response of the second-order Laplace transfer function (7).

Figure 11b,d show the evolutions, respectively, of the rotor axial deviation and axial force in response to the stepped axial deviation $\Delta z = -10$ mm. The evolutions of Figure 11d were determined from the ones in Figure 11b, using functions fitting the experimental characteristic presented in Figure 9a. The evolutions obtained from experimental measurements are in continuous blue lines. The dotted red line corresponds to the third-order Laplace transfer function's response (8).

$$G_r(s) = \frac{1773.1}{s^2 + 1.0188\,s + 1746.2} \tag{7}$$

$$G_a(s) = \frac{23932}{s^3 + 31.9\,s^2 + 1223.7\,s + 24076} \tag{8}$$

The transfer functions (7) and (8) correspond to the models that characterize with good approximation, respectively, rotor D5 vertical and rotor D20 axial dynamics. With these experimental results, the axial stability was demonstrated for the axial deviation of $\Delta z = -10$ mm in the D20 topology.

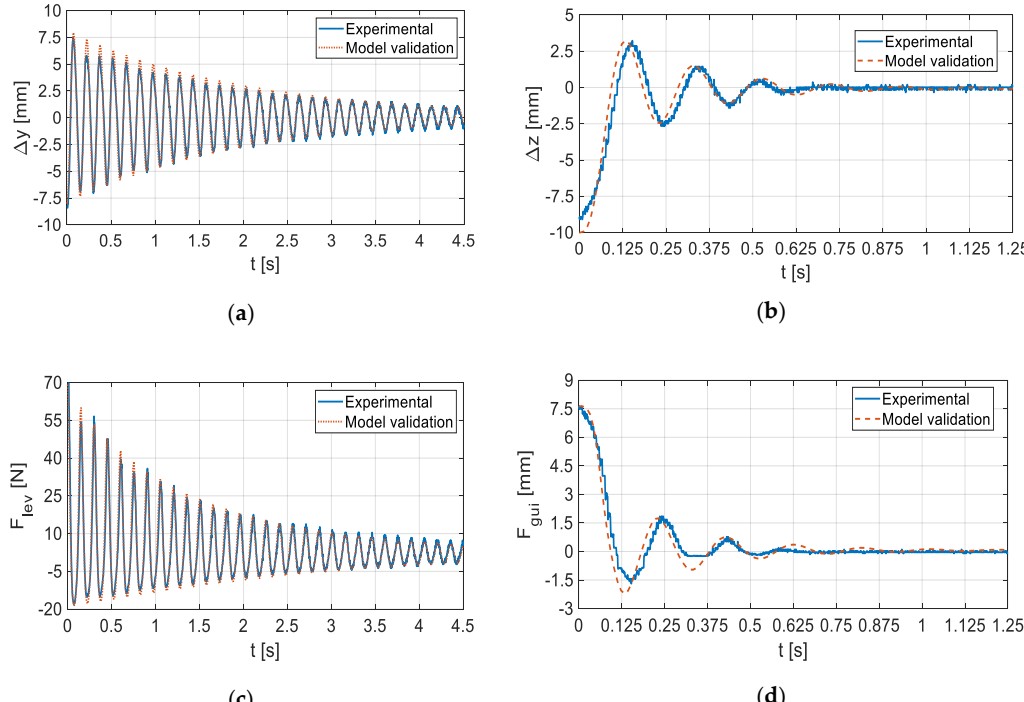

**Figure 11.** Evolutions of: (**a**) Rotor D5 vertical deviation; (**b**) Rotor D20 axial deviation; (**c**) Levitation force with Rotor D5 and the (**d**) guidance force with Rotor D20. Those are the dynamic responses to stepped vertical and axial deviations respectively of Rotors D5 and D20.

## 5. Optimization of the HTS and PM Ring Spacing to Maximize the Guiding Stability

A previous optimization to minimize the volume and cost of the 3D geometry and simultaneously maximize the net levitation force with a given constraint on the minimum net levitation force was performed in [6]. In this previous study, the component dimensions and spacing were all considered decision variables. The dimensions of the bulks used in Stator II are close to those indicated by the volume optimization.

This study aims to optimize the spacing between the two rings of HTS bulks in the stator and between the PM rings in the rotor, with the given bulk and PM ring sizes. This maximizes the maximum guidance force with a given constraint on the minimum net levitation force. Bulk sizes with volumes $33 \times 33 \times 14$ mm$^3$ (Stator I) and $40 \times 40 \times 10$ mm$^3$ (Stator II) are considered in this study. The considered PM ring dimensions are equal to the ones used in the bearing experimental prototypes built.

### 5.1. Optimization Methodology

Optimizations were performed for the geometries with six HTS bulks at the bottom of the stator. In 3D FEA, simulations were considered, and the stator and rotor axes were aligned. Several rotor axial deviations were performed for each genetic code (geometry solution) to find the maximum guiding force. Because of the existing symmetry, only 3/16 of the complete geometry was simulated to reduce the amount of numerical processing. Figure 12a,b show the 3D perspective, respectively, of the geometry and its simulated partition for the case with six bulks of volume $33 \times 33 \times 14$ mm$^3$ at the bottom of Stator I. Figure 13a,b refer to the case of bulks with volume $40 \times 40 \times 10$ mm$^3$.

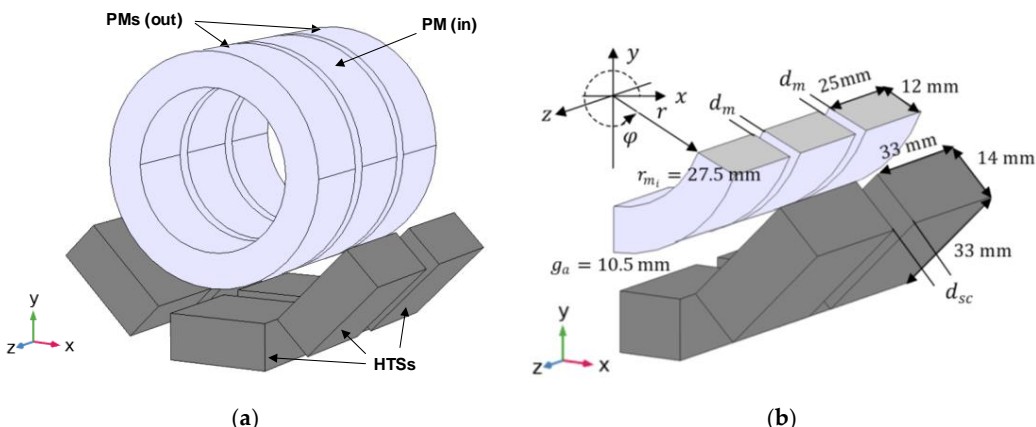

**Figure 12.** The 3D perspective of the (**a**) geometry and (**b**) simulated partition for the case of Stator I.

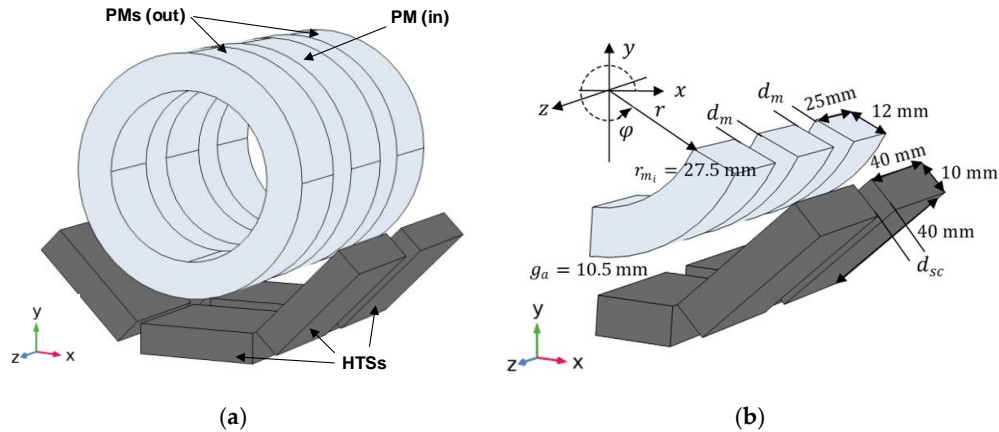

**Figure 13.** The 3D perspective of the (**a**) geometry and (**b**) simulated partition for the case of Stator II.

Optimizations were performed using the equivalent permeability model and not the E-J model to significantly reduce the numerical processing time required in the 3D FEA. With better thermal insulation conditions than those of the existing experimental prototypes, levitation forces would be closer to the ones predicted using the E-J model with $J_{c0} = 8 \times 10^7$ Am$^{-2}$. According to Table 2, resuming the levitation force values with the rotor axis at the center position, the values predicted with $\overline{\mu_r} = 0.25$ are close to the ones predicted with $J_{c0} = 8 \times 10^7$ Am$^{-2}$. For these reasons, 3D FEA optimization simulations run with $\overline{\mu_r} = 0.25$.

Decisions were taken using NSGA-II over 3D FEA results with $\overline{\mu_r} = 0.25$. This optimization procedure was already applied in [6] and verified experimentally. As decision variables, the distance $d_{sc}$ was chosen between the rings of HTS bulks in the stator and the distance $d_m$ between the PM rings in the rotor.

Because of the difficulty in physically implementing a rotor with spacing between PM rings lower than 5 mm, a minimum spacing between PM rings equal to the one in continuous PM rotor D5 was considered. The construction of a cryostat with walls made of polyurethane, having a minimum required thickness of 5 mm because of thermal insulation reasons, implies a minimum spacing between rings of HTS bulks of 10 mm. The maximum spacing between rings of HTS bulks was about 20% higher than the width of PM rings. The maximum spacing between PM rings was also about 20% higher than the width of HTS rings.

Table 3 states the ranges of decision variables considered in optimizing the spacing between rings of HTS bulks and PM rings.

**Table 3.** Ranges for decision variables in the optimization of spacing.

| Decision Variable | Description | Range [mm] |
|---|---|---|
| $d_m$ | Spacing between PM rings with Stator I | 5–40 |
| $d_m$ | Spacing between PM rings with Stator II | 5–50 |
| $d_{sc}$ | Spacing between rings of HTS bulks | 10–30 |

This optimization's objective function consisted of, is optimization's objective function consisted of maximizing the maximum guidance force and the net levitation force. Constraints on the minimum acceptable net levitation force are applied. The expressions in (9) traduce the adopted optimization criteria.

$$f_o = \left\{ Max \left( F_{gui_{max}} \right) ; \ Max \left( F_n \right) \right\} \quad ; \quad F_n \geq F_{n_{min}}. \tag{9}$$

The optimizations were made with a population size of 150 and for 50 generations. To find the maximum guidance force $F_{gui_{max}}$ and the corresponding axial deviation $\Delta_{z_{max}}$, ten equally spaced axial deviations $\Delta z$ of the rotor, considering the rotor and stator axes aligned, were simulated for each genetic code geometry. A limit sweep displacement $\Delta_{z_{lim}}$ given by Equation (10), was imposed to guarantee that the center of the middle PM ring does not surpass the centers of each HTS, as represented in Figure 14.

$$\Delta_z = i \frac{\Delta_{z_{lim}}}{10} \quad ; \quad i = 1, 2, \ldots, 10 \quad ; \quad \Delta_{z_{lim}} = \frac{w_{sc} + d_{sc}}{2}. \tag{10}$$

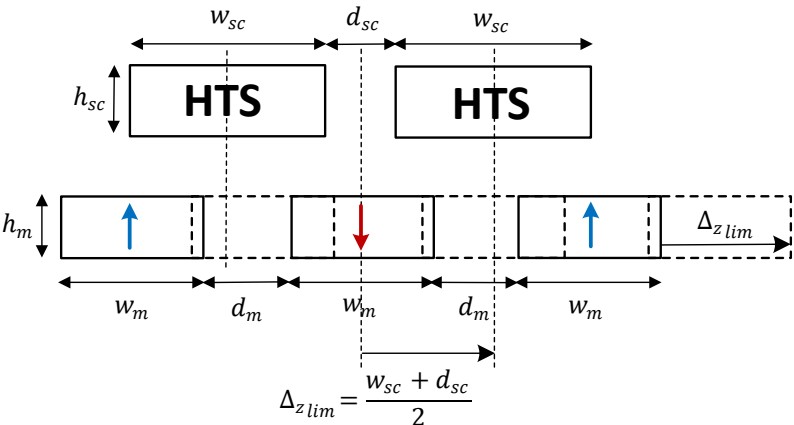

**Figure 14.** Schematic representation of the limit sweep displacement $\Delta_{z_{lim}}$ used in the determination of $F_{gui_{max}}$ and corresponding $\Delta_{z_{max}}$.

The value of $\Delta_{z_{lim}}$ is higher than $\Delta_{z_{max}}$ obtained when the center of the space between adjacent PMs with opposite polarization is aligned with the center of one HTS bulk as shown in Figure 14.

### 5.2. Optimization Results

Figure 15 shows the Pareto's fronts obtained for the case with six bulks of volume $33 \times 33 \times 14$ mm$^3$ at the bottom of Stator I. As expected, the maximum levitation force is verified for the genetic code with $d_m = 5$ mm and $d_{sc} = 10$ mm, pointed out by the left dashed line, corresponding to the geometry with Rotor D5 in Section 3. The genetic code with $d_m = 20$ mm and $d_{sc} = 10$ mm, corresponding to the geometry with Rotor D20 in Section 3, is pointed by the right dashed line.

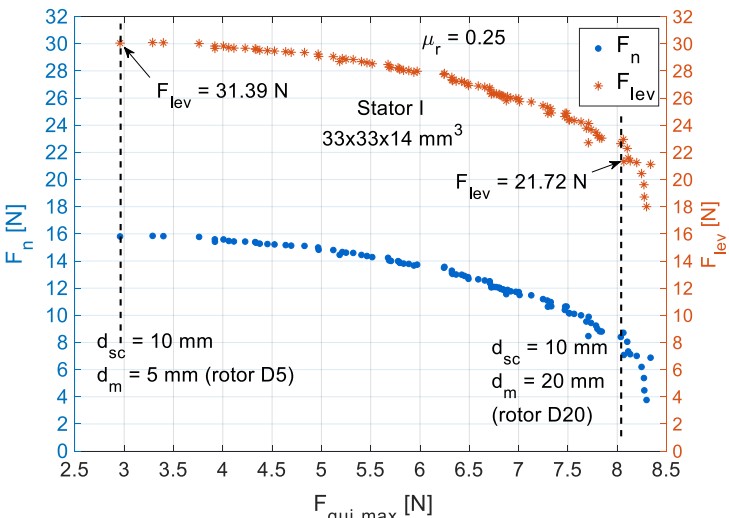

**Figure 15.** Pareto's front with six HTS bulks of $33 \times 33 \times 14$ mm$^3$ at the bottom of Stator I.

Figure 16 shows the Pareto's front obtained for the case with six bulks of volume $40 \times 40 \times 10$ mm$^3$ at the bottom of Stator II.

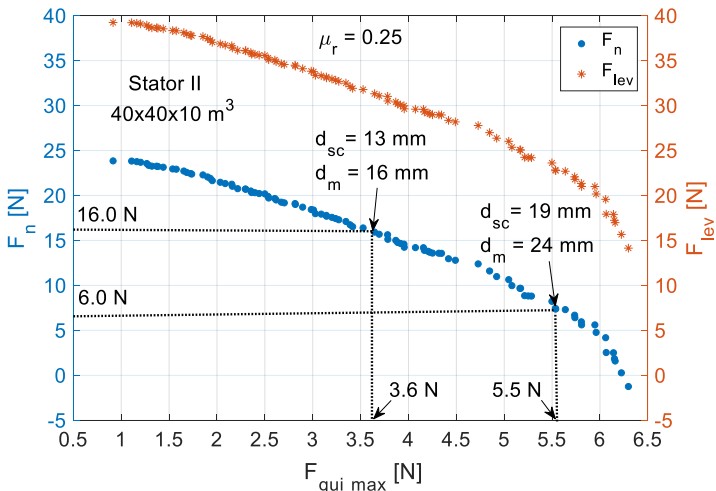

**Figure 16.** Pareto's front with six HTS bulks of $40 \times 40 \times 10$ mm$^3$ at the bottom of Stator II.

The geometry with bulks of volume $40 \times 40 \times 10$ mm$^3$, with $d_m = 16$ mm and $d_{sc} = 13$ mm, guarantees the same minimum levitation force as the restriction geometry with bulks of volume $33 \times 33 \times 14$ mm$^3$, with $d_m = 5$ mm (Rotor D5) and $d_{sc} = 10$ mm, for which the levitation and guidance force characteristics were presented in Section 3. With this geometry, the maximum guidance force is $F_{gui_{max}} = 3.6$ N. while for the geometry with bulks of volume $33 \times 33 \times 14$ mm$^3$, $d_m = 5$ mm and $d_{sc} = 10$ mm, the maximum guidance force was $F_{gui_{max}} = 3.1$ N.

The geometry with bulks of volume $40 \times 40 \times 10$ mm$^3$, with $d_m = 24$ mm and $d_{sc} = 19$ mm, guarantees the same minimum levitation force as the geometry with bulks of volume $33 \times 33 \times 14$ mm$^3$, $d_m = 20$ mm (Rotor D20) and $d_{sc} = 10$ mm, for which levitation and guidance force characteristics were also presented in Section 3. With this geometry, the maximum guidance force is $F_{gui_{max}} = 5.5$ N while for the geometry with bulks of volume $33 \times 33 \times 14$ mm$^3$, $d_m = 20$ mm and $d_{sc} = 10$ mm, the maximum guidance force was $F_{gui_{max}} = 8.24$ N.

Figure 17 shows the distribution of **B** and **J** obtained by 3D FEA using the E-J model with $J_{c0} = 8 \times 10^7$ Am$^{-2}$, for the optimized geometry with six bulks of $40 \times 40 \times 10$ mm$^3$

in Stator II, $d_m = 16$ mm and $d_{sc} = 13$ mm, providing the same net levitation as with six bulks of $33 \times 33 \times 14$ mm$^3$ at the bottom of Stator I and Rotor D5.

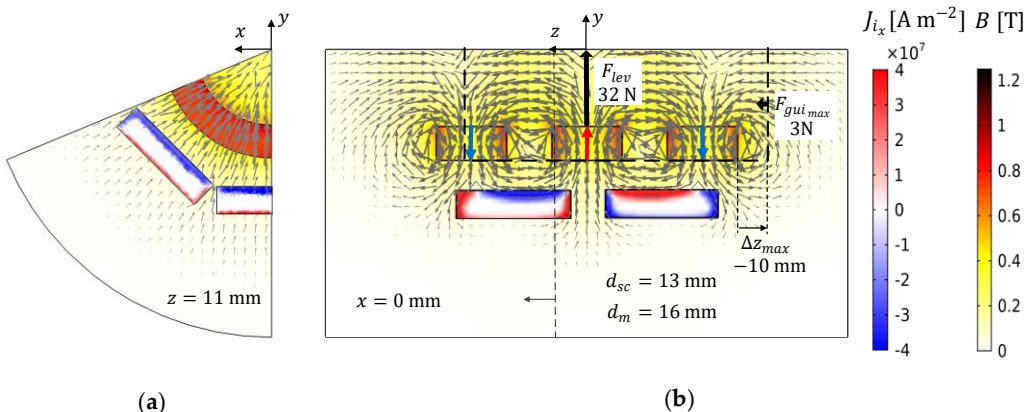

(**a**)                                        (**b**)

**Figure 17.** (**a**) Transversal and (**b**) longitudinal views with the distributions of **B** and **J** for the optimized geometry with bulks of volume $40 \times 40 \times 10$ mm$^3$, $d_m = 16$ mm and $d_{sc} = 13$ mm.

Figure 18 shows the distribution of **B** and **J** obtained by 3D FEA using the E-J model with $J_{c0} = 8 \times 10^7$ Am$^{-2}$, for the optimized geometry with six bulks of $40 \times 40 \times 10$ mm$^3$, $d_m = 24$ mm and $d_{sc} = 19$ mm, providing the same net levitation as with six bulks of $33 \times 33 \times 14$ mm$^3$ at the bottom of Stator I and Rotor D20.

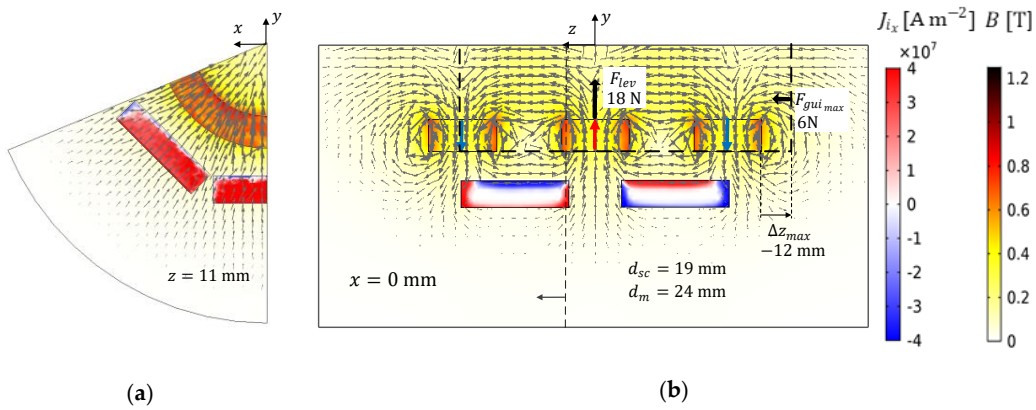

(**a**)                                        (**b**)

**Figure 18.** (**a**) Transversal and (**b**) longitudinal views with the distributions of **B** and **J** for the optimized geometry with bulks of volume $40 \times 40 \times 10$ mm$^3$, $d_m = 24$ mm and $d_{sc} = 19$ mm.

Diagrams showing the vectors $F_{lev}$ and $F_{gui_{max}}$ are represented in Figures 17b and 18b. The dashed lines represent the rotor position at the axial deviation $\Delta z_{max}$ for which the guidance force magnitude is the maximum. For the second geometry, $F_{gui_{max}}$ is higher and $F_{lev}$ is lower than in the first geometry.

To validate optimization results with Stator II, the measurement of several combinations of distances $d_m$ and $d_{sc}$ were performed with this stator. Plastic bars, 3 mm thick, were inserted between the two rings of HTS bulks in Stator II to control $d_{sc}$. Plastic washers, 2.5 mm thick, were inserted between the PM rings to control $d_m$.

Experimental measurements were performed for the case where $d_m = 17.5$ mm and $d_{sc} = 13$ mm. In this case, five washers of 2.5 mm thick were inserted between the PM rings in addition to the original 5 mm disks of Rotor D5, as shown in Figure 19a. One 3 mm thick plastic bar was inserted between the two chambers of Stator II, in addition to their minimum distance of 10 mm, as shown in Figure 19b. Figure 19c shows the measurements of $F_n$ and $F_{lev}$ with no vertical deviation of the rotor from the central position. Figure 19d shows the measurement of $F_{gui_{max}}$.

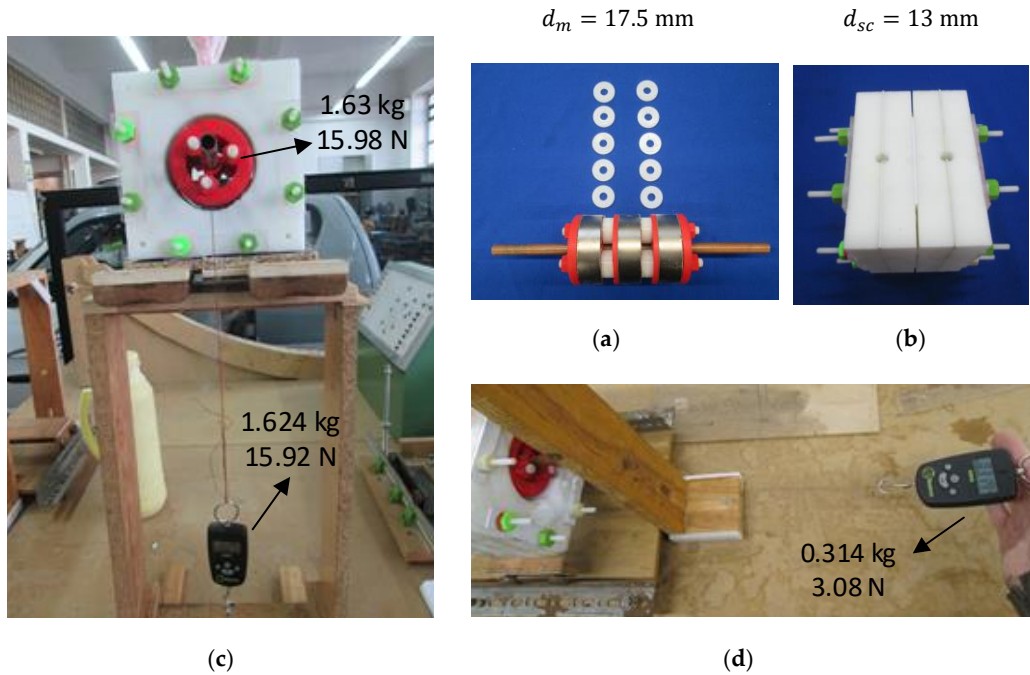

$d_m = 17.5$ mm          $d_{sc} = 13$ mm

(a)          (b)

(c)          (d)

**Figure 19.** (**a**) Rotor for $d_m = 17.5$ mm, (**b**) Stator II for $d_{sc} = 13$ mm, (**c**) measurement of $F_n$ and $F_{lev}$ with no vertical deviation of the rotor, and (**d**) measurement of $F_{gui_{max}}$.

Figure 20a,b show the characteristics of $F_{lev}$ and $F_{gui}$ by 3D FEA with $\overline{\mu_r} = 0.25$ and $J_{c0} = 8 \times 10^7$ Am$^{-2}$, plotting the experimental measurements of $F_{lev}$ with $\Delta_y = 0$ and of $F_{gui_{max}}$, with $d_{sc} = 13$ mm, and $d_m = 15$ mm or $d_m = 17.5$ mm. The experimental value of $F_{lev}$ in Figure 20a is the sum of the net force $F_n = 15.92$ N, given by the dynamometer with the gravity force of the rotor weight $F_n = 15.98$ N. As one may verify, experimental values are close to the ones predicted by 3D FEA.

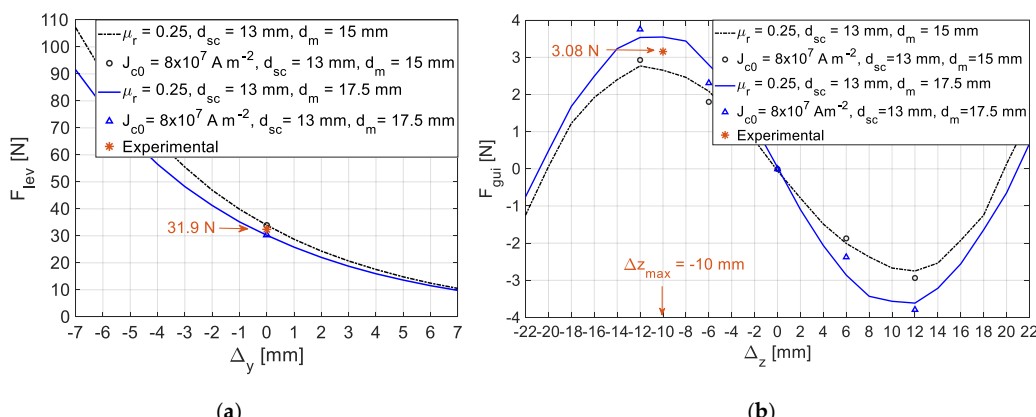

(a)          (b)

**Figure 20.** 3D FEA and experimental results for (**a**) $F_{lev}$, and (**b**) $F_{gui}$ with $d_{sc} = 13$ mm, and $d_m = 15$ mm or $d_m = 17.5$ mm.

Experimental measurements were also performed with $d_m = 25$ mm and $d_{sc} = 19$ mm. In this case, two 2.5 mm thick washers were inserted between the PM rings in addition to the original 20 mm disks of Rotor D20, as shown in Figure 21a. Three 3 mm thick plastic bars were inserted between the two chambers of Stator II, in addition to the minimum distance of 10 mm, as in Figure 21b.

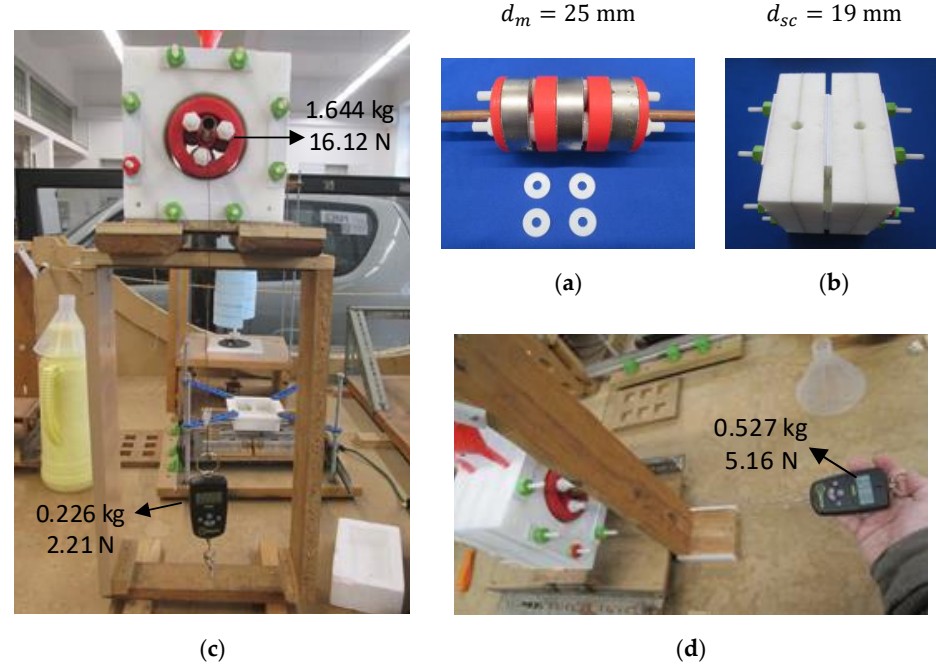

$d_m = 25$ mm $\qquad$ $d_{sc} = 19$ mm

(a) $\qquad$ (b)

(c) $\qquad$ (d)

**Figure 21.** (**a**) Rotor for $d_m = 25$ mm, (**b**) Stator II for $d_{sc} = 19$ mm, (**c**) measurement of $F_n$ and $F_{lev}$ with no vertical deviation of the rotor, and (**d**) measurement of $F_{gui_{max}}$.

Figure 22a,b show the $F_{lev}$ and $F_{gui}$ characteristics obtained by 3D FEA with $\overline{\mu_r} = 0.25$ and $J_{c0} = 8 \times 10^7$ Am$^{-2}$, plotting the experimental measurements of $F_{lev}$ with $\Delta_y = 0$ and of $F_{gui_{max}}$. This is for the combinations with $d_{sc} = 19$ mm and $d_m = 22.5$ mm or $d_m = 25$ mm.

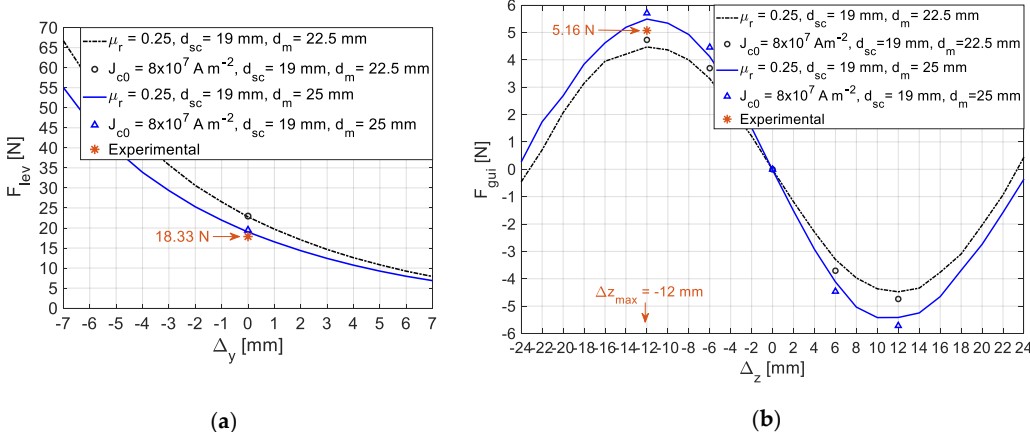

(a) $\qquad$ (b)

**Figure 22.** 3D FEA and experimental results for (**a**) $F_{lev}$, and (**b**) $F_{gui}$ with $d_{sc} = 19$ mm, and $d_m = 22.5$ mm or $d_m = 25$ mm.

The experimental value of $F_{lev}$ in Figure 22a is the sum of the net force $F_n = 2.21$ N given by the dynamometer with the gravity force of the rotor weight $F_n = 16.12$ N. The value of $F_{gui_{max}}$ is higher for the geometry with six bulks of $33 \times 33 \times 14$ mm$^3$ at the bottom at Stator I and Rotor D20.

This section presented the obtained Pareto's fronts for the simultaneous maximization of the net levitation force $F_n$ and maximum guidance forces $F_{gui_{max}}$ for the cases of six bulks of $33 \times 33 \times 14$ mm$^3$ in Stator I and six bulks of $40 \times 40 \times 10$ mm$^3$ in Stator II. From the two Pareto curves, one may verify that the higher the $F_{gui_{max}}$ the lower the guaranteed $F_n$ and $F_{lev}$. This is because, as more magnetic flux and energy are used for levitation, less magnetic flux and energy are used for guidance. From optimization results with bulks

of $40 \times 40 \times 10$ mm$^3$, two combinations of spacing $d_m$ and $d_{sc}$ are derived that maximize $F_{gui_{max}}$, each providing the same net levitation as with Rotor D5 or Rotor D20 and six bulks of $33 \times 33 \times 14$ mm$^3$ at the bottom of Stator I.

The values of $F_{gui_{max}}$ and $F_{lev}$ predicted for the two optimized geometries with bulks of $40 \times 40 \times 10$ mm$^3$ were validated experimentally. From the several combinations tested, it was verified that it is only possible to keep the rotor in a stable position when the distance of the far end edges of the side PM rings in the rotor is higher than the distance of the far end edges of the two HTS bulk rings in the stator.

## 6. Identification of Guiding Stability Zones

Several 3D FEA simulations using $\overline{\mu_r} = 0.25$, ran for spacings $d_{sc}$ from 10 mm to 25 mm with steps of 3 mm equal to the thickness of the plastic bars inserted between the two rings of HTS bulks in Stator II, and for spacing $d_m$ from 5 mm to 25 mm with steps of 2.5 mm equal to the thickness of the washers inserted between PM rings.

Figure 23a,b show the surfaces of $F_{lev}$ for Stator I and Stator II.

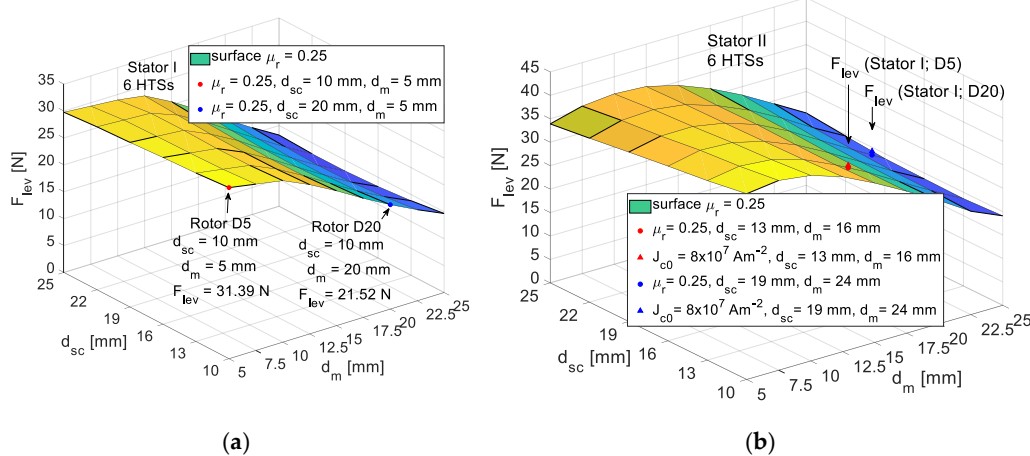

**Figure 23.** Surfaces of dependence of $F_{lev}$ from $d_{sc}$ and $d_m$ for (**a**) Stator I, and (**b**) Stator II.

As one may verify, the values of $F_{lev}$ are generally higher for the case of Stator II, including bulks of volume $40 \times 40 \times 10$ mm$^3$.

Figure 24a,b show the surfaces of $F_{gui_{max}}$ for Stator I and Stator II.

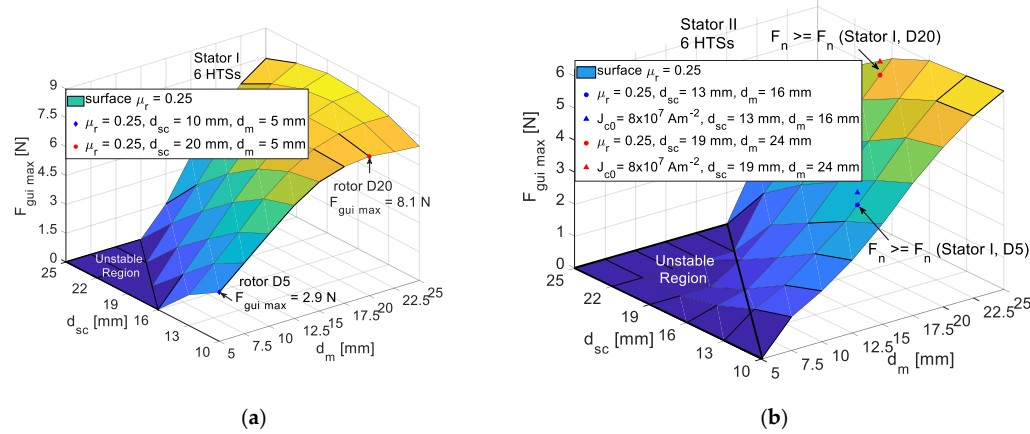

**Figure 24.** Surfaces of dependence of $F_{gui_{max}}$ from $d_{sc}$ and $d_m$ for (**a**) Stator I, and (**b**) Stator II.

As one may verify with bulks of volume $33 \times 33 \times 14$ mm$^3$ (Stator I), the values of $F_{gui_{max}}$ are generally higher than with bulks of volume $40 \times 40 \times 10$ mm$^3$ (Stator II). The region of no guiding stability is also wider for the last case (Stator II).

Figure 25 shows the distribution of magnetic flux and current densities obtained by 3D FEA using the E-J model with $J_{c0} = 8 \times 10^7$ Am$^{-2}$, for one geometry on the limit of the region of guiding stability with $d_{sc} = 19$ mm and $d_m = 10$ mm.

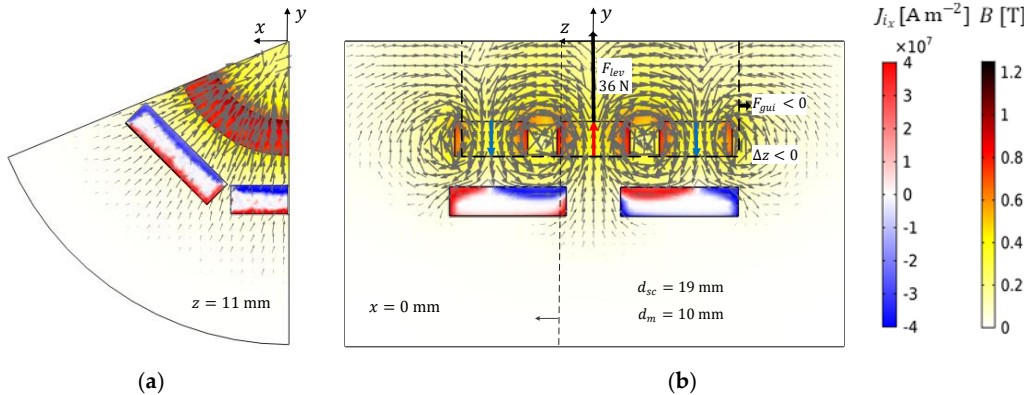

(**a**)　　　　　　　　　　　　　　　　　　(**b**)

**Figure 25.** (**a**) Transversal and (**b**) longitudinal views with the distributions of **B** and **J** for the geometry with bulks of volume $40 \times 40 \times 10$ mm$^3$, $d_m = 10$ mm and $d_{sc} = 19$ mm.

As one may verify, the bulk extremities are almost coincident with the extremities of the two side PM rings for this geometry. This generally occurs for the geometries on the limit line of guiding stability. Thus, the condition that should be considered for an initial check of the existence of guiding stability is the one expressed by (11),

$$\frac{d_{sc}}{2} + w_{sc} \le d_m + 3\frac{w_m}{2}, \tag{11}$$

from which the minimum limit for $d_m$ may be deduced, given by (12),

$$d_m \ge \frac{d_{sc}}{2} - 3\frac{w_m}{2} + w_{sc}. \tag{12}$$

Table 4 presents the minimum values of $d_m$ for several values of $d_{sc}$, on the cases of Stator I and Stator II, calculated using (12). The values in Table 4 approximately follow the lines on the limit of the guiding stability regions in Figure 24a,b.

**Table 4.** Minimum values of $d_m$ for several values of $d_{sc}$, on the cases of Stator I and Stator II.

| | Minimum $d_m$ [mm] | |
|---|---|---|
| $d_{sc}$ [mm] | Stator I | Stator II |
| 10 | 0.5 | 7.5 |
| 13 | 2 | 9 |
| 16 | 3.5 | 10.5 |
| 19 | 5 | 12 |
| 22 | 6.5 | 13.5 |
| 25 | 8 | 15 |

As verified experimentally, if the bulk extremities become narrower than the PM extremities of the two side PM rings, the guidance forces assume negative or positive values respectively for negative or positive axial deviations, and no guiding stability is guaranteed.

## 7. Conclusions

In this study, geometries that maximize the guiding stability of a horizontal axis radial levitation bearing with HTS bulks cooled by ZFC were determined.

Two different stators for two different bulk sizes ($33 \times 33 \times 14$ mm$^3$ and $40 \times 40 \times 10$ mm$^3$) were studied. The domain of combinations of the spacing $d_{sc}$ between the two HTS bulk rings and the spacing $d_m$ between PM rings, where there is no guiding stability, were determined for the two stator cases. The domain where there is no guiding stability is higher for the case of $40 \times 40 \times 10$ mm$^3$ size bulks in Stator II than for the case of $33 \times 33 \times 14$ mm$^3$ size bulks in Stator I.

The optimization of spacing performed for the two stator cases resulted in Pareto fronts, from where one can obtain the maximum possible value of $F_{gui_{max}}$, for a given restriction on the minimum levitation force $F_{lev}$ that should be guaranteed. In the Pareto front for bulks of $33 \times 33 \times 14$ mm$^3$, the possible maximum guidance force is higher. With six bulks of $33 \times 33 \times 14$ mm$^3$ at the bottom of Stator I and Rotor D20, the maximum guidance is about $F_{gui_{max}} = 8$ N. Once $\Delta_{z_{max}} = 10$ mm, the corresponding maximum available energy to pull back the rotor to equilibrium is about 0.08 J. The corresponding guiding stiffness is about $k_z = 800$ Nm$^{-1}$ for the geometry with six bulks of $40 \times 40 \times 10$ mm$^3$ at the bottom of Stator II, with optimized spacing $d_{sc} = 19$ mm and $d_m = 24$ mm, which guarantees the same net levitation force as with Stator I and Rotor D20; the maximum guidance is only about $F_{gui_{max}} = 5.5$ N. In this case, $\Delta_{z_{max}} = 12$ mm, and the maximum available energy to pull back the rotor is about 0.066 J. The corresponding guiding stiffness is about $k_z = 458.3$ Nm$^{-1}$. Thus, with bulks of $33 \times 33 \times 14$ mm$^3$ in Stator I, the maximum guidance force is generally higher than with bulks of $40 \times 40 \times 10$ mm$^3$ in Stator II.

To have guiding stability, the length between the edges of the side PM rings in the rotor should be higher than the length between the edges of the two HTS bulk rings in the stator. From this conclusion, a condition that should be considered in the bearing geometry design was driven. From the experimental results with six bulks of $40 \times 40 \times 10$ mm$^3$ at the bottom of Stator II, the measured maximum guidance force was about $F_{gui_{max}} = 5.16$ N with $d_{sc} = 19$ mm and $d_m = 25$ mm and only about $F_{gui_{max}} = 3.08$ N with $d_{sc} = 13$ mm and $d_m = 17.5$ mm. The difference in the length between the edges of the side PMs rings and the length between the edges of the two bulk rings is 26 mm on the first case and 17 mm on the second case. From these results, one may conclude that the guiding stability is higher in the first case.

**Author Contributions:** Conceptualization, P.J.C.B.; investigation, A.J.A., J.F.P.F., F.F.d.S., P.J.C.B.; writing—original draft preparation, A.J.A., J.F.P.F., F.F.d.S.; writing—review and editing, A.J.A., J.F.P.F., F.F.d.S., P.J.C.B.; supervision, P.J.C.B.; project administration, P.J.C.B. All authors have read and agreed to the published version of the manuscript.

**Funding:** This work was supported by FCT, through IDMEC, under LAETA, projects UID/EMS/50022/2013, PTDC/EEEIEEL/4693/2014—HTSISTELEC, and UIDB/50022/2020, and by FCT fellowship SFRH/BD/117921/2016 to A.J. Arsénio as a Ph.D. student in IST.

**Acknowledgments:** With the support of FabLabs from EDP and Lisbon's City Hall, in CNC milling machine works during the construction of the bearing prototypes' stators.

**Conflicts of Interest:** The authors declare no conflict of interest.

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
