# Peer review of "Optimization of the Guiding Stability of a Horizontal Axis HTS ZFC Radial Levitation Bearing"

_actuators, doi:10.3390/act10120311_

Round 1
Reviewer 1 Report
This paper is well documented on techniques for optimizing the stability of horizontal axes for HTS ZFC Radial Bearings.
I will comment below.
In Introduction, there are many citations of papers from own institution, and few reports presented by other institutions. In addition, in order to clarify the uniqueness and effectiveness of this paper, it is necessary to express advantages of this research while citing the research of other institutions. In addition, it is necessary to add details of why stability optimization is necessary more deeply.
In optimization, it is important to understand the accuracy of FEA analysis and the tendency of calculation results. Clearly showing the difference in analysis accuracy and output result tendency between each calculation model, such as constant permeability model and E-J model will be needed. In addition, consideration of the difference in the tendency of analysis results(validation with experimental values) would be required. And then, describing the advantages and limits of each model can give the reader a deep understanding.
A detailed description of the experimental equipment and method is required. For example, how did you displace the floater? How did you measure the force accurately? What is the measurement accuracy of displacement and force? Must be described.
It is difficult to understand the relationship between the variable parameters and the analysis results. It is necessary to clearly describe the relationship between the design parameter and the result.
Quantitative results are described for the results section. In contrast, discussion for the results and explanation of the phenomena/tendency is less discribed. Discussion for results should be added to give the validity and correctness of the results to readers.
Author Response
Point 1: In Introduction, there are many citations of papers from own institution, and few reports presented by other institutions. In addition, in order to clarify the uniqueness and effectiveness of this paper, it is necessary to express advantages of this research while citing the research of other institutions. In addition, it is necessary to add details of why stability optimization is necessary more deeply.
Response 1: In page 4 were inserted references to other work and highlight the advantages of this research. The studied frictionless bearing can be applied in high-speed applications. Induction currents due to dynamics, could appear even with ZFC. Active control to generate compensation forces and reduce vibrations and losses could by necessary for high-speed applications. References [7,8] were included because present studies comparing the levitation force with ZFC and with FC at several heights, where it is shown that the levitation force with ZFC is higher. Reference [9] presents a study on levitation and guidance forces of a Maglev launch assist test vehicle, of similar geometry, including a guideway with three lines of PMs in alternate aligned configuration and HTS bulks. The results show that with ZFC the obtained levitation gap is higher than with FC at several heights. Presents also results on the guidance force depending on the lateral displacement. References [10,11] present a multi-objective geometry optimization of a bearing topology similar to the one of the studied bearing. Reference [12] presents a methodology to determine induction current losses (AC losses), which are the ones that mainly could occur with ZFC. Reference [13] presents a study on the geometry optimization of axial flux machine to reduce the torque ripple. Finally, reference [15] highlights the importance of active generation of compensation forces to control the displacement of a flywheel.
The advantages of this research are related with the adoption of ZFC to minimize the magnetization energy and Joule losses in bulks to maximize their lifetime. The other reason is that with ZFC impulsion forces are higher than with FC at several heights. With ZFC there is no initial flux pinning and stability is not guaranteed as with FC. Hence, a specific bearing geometry with a specific arrangement of PMs and distribution of HTS bulks is used to promote creation of guidance forces with ZFC. This study complements the research done on this bearing, by optimizing the spacings between PMs rings and rings of HTS bulks to maximize the guidance force and the guiding stability. This explanation, expressed in the last four paragraphs of page 4, details why stability optimization is necessary for this ZFC bearing.
Point 2: In optimization, it is important to understand the accuracy of FEA analysis and the tendency of calculation results. Clearly showing the difference in analysis accuracy and output result tendency between each calculation model, such as constant permeability model and E-J model will be needed. In addition, consideration of the difference in the tendency of analysis results (validation with experimental values) would be required. And then, describing the advantages and limits of each model can give the reader a deep understanding.
Response 2: The first paragraph of page 7 was inserted to detail about resolution of the mesh and interpolation method used on FEA. A study of the accuracy of FEA analysis when a linear shape function is adopted, depending on the mesh resolution and on the electromagnetic model, was done in reference [22], cited in this paragraph. The software used for FEA is now cited by reference [23].
Point 3: A detailed description of the experimental equipment and method is required. For example, how did you displace the floater? How did you measure the force accurately? What is the measurement accuracy of displacement and force? Must be described.
Response 3: The description of the experimental equipment used and methodology for measurement of the levitation force depending on the vertical deviation and of axial force depending on the axial deviation keeping the rotor and stator axes aligned is now performed in last two paragraphs of page 7. In this description is explained how the vertical and axial deviations were precisely set.
Point 4: It is difficult to understand the relationship between the variable parameters and the analysis results. It is necessary to clearly describe the relationship between the design parameter and the result.
Response 4: Guidance forces are measured with the rotor and stator axes aligned. From Figures 8 and 9 one may verify that with the rotor and stator axes aligned the guidance force is not so sensible to the value of equivalent permeability as the levitation force. Table 2 compares the levitation forces predicted by the equivalent permeability model using values of mur between 0.2 and 0.4 or by the E-J model using values of Jc0 equal to 3x10^7 [A/m^2] and 8x10^7 [A/m^2] with the ones measured experimentally. The error is now stated for each model and parameter value. According to this table the error with the equivalent permeability model is minimized with mur=0.25 for the case of Rotor D5. During the set of experiences with rotor D5 and Stator I, this stator presented better thermal and liquid insulation conditions than during the set of experiences with Rotor D20. Hence, mur=0.25 is appropriate when the rotor and stator axes are aligned. This is explained in the last part of section 3 dedicated to the validation of electromagnetic model parameters.
The optimization methodology described in section 5.1 is based on these observations. During the optimization process, magnetic forces were predicted for all genetic codes (set of decision variables) with the rotor and stator axes aligned and using the equivalent permeability model with mur=0.25 to significantly reduce the numerical processing time.
Point 5: Quantitative results are described for the results section. In contrast, discussion for the results and explanation of the phenomena/tendency is less described. Discussion for results should be added to give the validity and correctness of the results to readers.
Response 5: Two last paragraphs were added at the end of the new section 5.2 to clarify readers about the validity, coherence and meaning of quantitative results. In the new section 7, the main conclusions are now supported by quantitative results from the new section 5.2.
Please see the attachment.

Reviewer 2 Report
The paper structure is complete and the data is credible.It is recommended that the conclusions of Part 5 be reorganized.Mainly to get the illustrated results, we suggest a qualitative and quantitative combination.
Author Response
Point 1: The paper structure is complete and the data is credible. It is recommended that the conclusions of Part 5 be reorganized. Mainly to get the illustrated results, we suggest a qualitative and quantitative combination.
Response 1: The new section 7 was extended to include quantitative results supporting the main conclusions of the document.
Please see the attachment.

Reviewer 3 Report
General remarks – main paper problems:
- For the domestic rotor AMB user more important is the AMB dynamics, not a static characterization. Therefore the next step for performed research should be a dynamic characterisation of the magnetic bearing, where electromagnetic force vs time, control currents and air-gap is derived.
- There is lack of the magnetic bearing electrical and mechanical parameters, in particular for electromagnets, PM, and geometrical dimensions (stator, rotor), which should be given in the table.
- Also in the case of FEM numerical analysis, there is no set-up data.
- According to the paper results, the presentation should be clear to the less familiar readers. The authors should provide a more comprehensible description that, I believe, will be beneficial for magnetic bearings designers and users.
- Also, the authors should comment on what efficiency is achieved according to the proposed configuration, in particular to the flux loss?
- The conclusions should be supported with results.
Author Response
Point 1: For the domestic rotor AMB user more important is the AMB dynamics, not a static characterization. Therefore the next step for performed research should be a dynamic characterisation of the magnetic bearing, where electromagnetic force vs time, control currents and air-gap is derived.
Response 1: A new section 4 was included in the document, where are presented results on the characterization of dynamics of this passive magnetic bearing PMB. These include experimental measurements of the vertical dynamics in response to a stepped variation on the rotor vertical deviation, for the case of the geometry with Rotor D5 and six bulk of 33x33x14mm^3 at the bottom of Stator I. From the measurement of the vertical position response was calculated the evolution the corresponding levitation force evolution, based on a function that fits the experimental characteristic of Figure 8(b).
Also, experimental measurements of the axial dynamics in response to a stepped variation on the rotor axial deviation, for the case of the geometry with Rotor D20 and six bulk of 33x33x14mm^3 at the bottom of Stator I. From the measurement of the axial position response was calculated the evolution the corresponding guidance force evolution, based on a function that fits the experimental characteristic of Figure 9(a).
Transfer functions of the models that approximately characterize the vertical and axial rotor dynamics were also determined and presented.
Point 2: There is lack of the magnetic bearing electrical and mechanical parameters, in particular for electromagnets, PM, and geometrical dimensions (stator, rotor), which should be given in the table.
Response 2: A new Table I was included in the introduction section, containing the electrical and mechanical parameters and main geometric dimensions (stator, rotor) of the bearing experimental prototypes used on the experimental validation of electromagnetic model parameters.
Point 3: Also in the case of FEM numerical analysis, there is no set-up data.
Response 3: Set-up data of FEM numerical analysis is now given in the first paragraph of page 7, between Figures 6 and 7. Details about resolution of the mesh and interpolation method used on FEA are given. A study of the accuracy of FEA analysis when a linear shape function is adopted, depending on the mesh resolution and on the electromagnetic model, was done in reference [22], cited in this paragraph. The software used for FEA is now cited by reference [23].
Point 4: According to the paper results, the presentation should be clear to the less familiar readers. The authors should provide a more comprehensible description that, I believe, will be beneficial for magnetic bearings designers and users.
Response 4: A revision was made with a more comprehensible description to clarify the presentation to the less familiar readers.
Point 5: Also, the authors should comment on what efficiency is achieved according to the proposed configuration, in particular to the flux loss?
Response 5: Efficiency could be evaluated both in the levitation and guidance force requirements. As shown in the Pareto curves from new Figures 15 and 16. Higher guidance forces are obtained for lower net levitation forces. This is because as more magnetic flux and energy is used for levitation less magnetic flux and energy is used for guidance. This is commented in the penultima paragraph of the new Section 5.2.
Point 6: The conclusions should be supported with results.
Response 6: The new section 7 was extended to include quantitative results supporting the main conclusions of the document.
Please see the attachment.

Reviewer 4 Report
Review on manuscript entitled “Optimization of the Guiding Stability of a Horizontal Axis HTS ZFC Radial Levitation Bearing”.
The work presents an interesting optimization problem with numerical modeling of a horizontal axis radial levitation bearing with high-temperature superconductor bulks in the stator and radially magnetized permanent magnets in the rotor. Modeling results are confirmed with experiments in real scale. Paper is well organized, results are well described and informative. Generally, paper is interesting to read.
I have some recommendations which I believe will improve little bit your work:
“Guiding Stability” must be introduced better, as numerical criterion or other.
It is expected to present optimization procedure in much deeper details, parameter limitations, constrains, population individuals, etc. Some tables summarizing that data will be useful.
Some specific problems in modeling could be considered in deeper e.g. eddy current losses in permanent magnets, torque/force ripples, dynamic and static stability, etc.
Table 1, needs error estimation.
Figure 12, left vertical axis – Newton is with small letter, must be [N].
Figure 14 and Figure 15 – colorbar legends of current density and magnetic flux density ore wrongly labeled. Text labels are directly transferred from Figures 5-6, where they are correct.
Measurements presented on Figure 16, don’t seems to be extremely accurate, some comments on measurement accuracy could be added. Same is for 3D printed parts, they are far from engineering manufacturing accuracy. Please discuss with few words achieved actual accuracy of your prototype and its testing.
Reference list is very short for a journal paper. I could recommend to add more relevant works on the complex problem that you have. HTC modeling, magnetic bearing, electromagnetic optimization three really huge topics. Some support could be found in:
Eddy current losses in permanent magnets estimation
https://doi.org/10.1109/ICELMACH.2018.8506782
Force/Torque ripples estimation and optimization
https://doi.org/10.3390/math9151738
Magnetic bearing stability
https://doi.org/10.3390/act8030057
https://doi.org/10.3390/en9121051
Author Response
Point 1: “Guiding Stability” must be introduced better, as numerical criterion or other.
Response 1: The term “Guiding Stability” is now introduced in the penultimate paragraph of the introduction section (Section 1), saying that this stability can be measured by the maximum available energy to pull-back the rotor to the equilibrium position or by the axial or guiding stiffness. In the conclusions (Section 7), quantitative values of this maximum energy to pull-back the rotor, and the axial or guidance stiffness are presented for the two optimized geometries with bulks of 40x40x10 mm^3 providing the same net levitation forces as with bulks 33x33x14 mm^3 and respectively with rotors D5 and D20.
Point 2: It is expected to present optimization procedure in much deeper details, parameter limitations, constrains, population individuals, etc. Some tables summarizing that data will be useful.
Response 2: Deeper details were added in the last part of section 5.1 to complement the explanation of the optimization procedure, thus including the identification of decision variables and range of values for the cases of the two stators with two different bulk sizes. This optimization procedure was already applied in [6] and verified experimentally. The new Table 3 was included to resume the decision variables and their range of values. The population size and number of generations used for which there is a good definition of the optimization Pareto’s is also indicated. The used objective function and used restriction are well defined by the new expressions (9). The determination of the axial deviation limit and axial deviation steps used in the iterative determination of maximum guidance force and corresponding deviation, are now well defined by (10) and the new Figure 14.
Point 3: Some specific problems in modeling could be considered in deeper e.g. eddy current losses in permanent magnets, torque/force ripples, dynamic and static stability, etc.
Response 3: Studies with procedures to account for these specific problems are now referred in the introduction (Section 1) by including references:
[12] (https://doi.org/10.1109/ICELMACH.2018.8506782);
[13] (https://doi.org/10.3390/math9151738);
[14] (https://doi.org/10.3390/act8030057);
[15] (https://doi.org/10.3390/en9121051)
Point 4: Table 1, needs error estimation.
Response 4: The new Table 2 (previous Table 1) includes now error estimation.
Point 5: Figure 12, left vertical axis – Newton is with small letter, must be [N].
Response 5: The new Figure 15 (previous Figure 12) was corrected to have [N] in the units of the net force on the vertical axis.
Point 6: Figure 14 and Figure 15 – colorbar legends of current density and magnetic flux density ore wrongly labeled. Text labels are directly transferred from Figures 5-6, where they are correct.
Response 6: The text labels of the color bar legends of the new Figure 17 (previous Figure 14) and new Figure 18 (previous Figure 15) were exchanged/twisted being now correct as in Figures 5-6.
Point 7: Measurements presented on Figure 16, don’t seems to be extremely accurate, some comments on measurement accuracy could be added. Same is for 3D printed parts, they are far from engineering manufacturing accuracy. Please discuss with few words achieved actual accuracy of your prototype and its testing.
Response 7: Labels with measurement values from the dynamometer, were added to the new Figures 19(c) (previous Figure 16(c)) and Figure 19(d) (previous Figure 16(d)). A short description on how the measurement of levitation force was done, is now included in the paragraph between Figure 19 and Figure 20. The calculated measurements of levitation and maximum guidance forces are now highlighted with labels respectively in Figure 20(a) and Figure 20(b). The same clarification with inclusion of labels of measurements was also included in Figures 21 and 22.
As explained in the text, with relation to the spacing between PM rings in the rotor, this was controlled by the inclusion of 2.5mm thick plastic washers in addition to the existing 5mm and 20mm thick dishes respectively of rotors D5 and D20. With relation to the spacing between rings of HTS bulks, this was controlled by the inclusion of 3mm thick plastic bars in addition to the minimum spacing of corresponding to the aggregation of the two chamber walls with 5mm width each one.
Point 8: Reference list is very short for a journal paper. I could recommend to add more relevant works on the complex problem that you have. HTC modeling, magnetic bearing, electromagnetic optimization three really huge topics. Some support could be found in:
Response 8: The reference list was increased by adding the following 11 (eleven) new references:
[7] (https://10.1109/TASC.2005.849636)
[8] (https://0.1109/TASC.2010.2086034)
[9] (https://10.1109/TASC.2006.870013)
[10] (https://10.1007/978-3-540-78490-6_10)
[11] (Print ISBN:978-3-8007-3166-4)
[12] (https://doi.org/10.1109/ICELMACH.2018.8506782);
[13] (https://doi.org/10.3390/math9151738);
[14] (https://doi.org/10.3390/act8030057);
[15] (https://doi.org/10.3390/en9121051)
[22] (https://doi.org/10.1088/1361-6668/ab9544)
[23] Comsol version 5.5.
Please see the attachment.

Round 2
Reviewer 1 Report
Thank you for correcting your manuscript.
The detailed description of experimental method, and additional consideration to support the results improve the quality of this paper.
Following are additional comments.
I think that the error of μR: 18.4% in Rotor D20 is generally not negligible accuracy. In this regard, it is better to clearly mention the reliability of the analysis.
Your experimental values ​​are well obtained, but it seems that the method of supporting the dynamometer by hand, using a wire, and visually recognizing the number of rotations of the nut to determine the displacement is deteriorate the accuracy and repeatability of performed experiment. The limitation of the experimental method compared to a rigid experimental system should be discussed if necessary.
The good addition of discussions is very effective to improve the quality of this paper.
In addition, since the magnetic flux density B and the current density J are visualized by FEA, it is better to clearly show the relationship between the obtained levitation force/guiding force and the result obtained from the vector diagram.
Author Response
Point 1: I think that the error of μR: 18.4% in Rotor D20 is generally not negligible accuracy. In this regard, it is better to clearly mention the reliability of the analysis.
Response 1: Additional comments were inserted in page 9 after Table 2, with relation to the experimental measurements with Rotor D20 and Rotor D5, explaining why those obtained with Rotor D5 were considered more reliable for the validation of mur=0.25, in the case of alignment of the rotor and stator axes.
“With the rotor and stator axes aligned, the levitation force predicted with Jc0=8x10^7 A/m^2 is close to the one predicted with mur=0.25. The experimental values with Rotor D5 are close to the ones predicted with Jc0=8x10^7 A/m^2 and mur=0.25. It was considered to select only one value of to represent both rotors D5 and D20. The rotor D5 experiments were made first, when the bulks and PUR container were at its best conditions. After some use, the thermal insulation of the PUR wall may slightly reduce, and also the HTS bulk may lower its critical current. Therefore, a value of equivalent relative permeability mur=0.25 was selected for the prediction of forces during the optimization process with the rotor and stator axes aligned.”
Point 2: Your experimental values ​​are well obtained, but it seems that the method of supporting the dynamometer by hand, using a wire, and visually recognizing the number of rotations of the nut to determine the displacement is deteriorate the accuracy and repeatability of performed experiment. The limitation of the experimental method compared to a rigid experimental system should be discussed if necessary. The good addition of discussions is very effective to improve the quality of this paper.
Response 2: An additional explanation was inserted in page 8 to explain the reasons of the methods used in experimental measurement of forces and detail about the accuracy of measurements.
“The experimental measurement of forces with a dynamometer and a tensor wire enabled the placing of the non-cryogenic dynamometer away from the stator avoiding its malfunctioning and damage by freezing. For a specific rotor vertical or axial deviation, a minimum of three levitation or guidance force measurements were done considering the mean value. Each reading was made after the rotor stopped oscillating relatively to the stator and the dynamometer display stopped scanning showing a fixed measurement.”
Point 3: In addition, since the magnetic flux density B and the current density J are visualized by FEA, it is better to clearly show the relationship between the obtained levitation force/guiding force and the result obtained from the vector diagram.
Response 3: Diagrams showing and relating the vectors of the levitation force F_lev with the rotor and stator axes aligned and no rotor axial deviation and the maximum guidance force F_gui_max obtained by pulling the rotor with a negative axial deviation keeping the rotor and stator axes aligned, were included in Figures 5(b), 6(b), 17(b), 18(b) and 25(b). Additional text was inserted on pages 7, 8 and 21 to explain these vector representations.
Note: A new version of the paper is submitted with both major and minor revisions in track changes. Minor revisions are highlighted with grey background color.
Reviewer 3 Report
No further comments.
Author Response
Point: No further comments.
Response: We appreciate that our responses to major revision comments were accept and no further comments were added for the minor revision.
Reviewer 4 Report
In my opinion, after the first round of revision, this manuscript is ready for publication.
Good work.
Author Response
Point: In my opinion, after the first round of revision, this manuscript is ready for publication.
Response: We appreciate that our responses to major revision comments were accept and no further comments were added for the minor revision.